# Hierarchical Reinforcement Learning with Advantage-Based Auxiliary Rewards

**Siyuan Li**[*]
IIIS, Tsinghua University
sy-li17@mails.tsinghua.edu.cn

**Rui Wang**[*]
Tsinghua University
rui1@stanford.edu

**Minxue Tang**
Tsinghua University
tangmx16@mails.tsinghua.edu.cn

**Chongjie Zhang**
IIIS, Tsinghua University
chongjie@tsinghua.edu.cn

## Abstract

Hierarchical Reinforcement Learning (HRL) is a promising approach to solving long-horizon problems with sparse and delayed rewards. Many existing HRL algorithms either use pre-trained low-level skills that are unadaptable, or require domain-specific information to define low-level rewards. In this paper, we aim to adapt low-level skills to downstream tasks while maintaining the generality of reward design. We propose an HRL framework which sets auxiliary rewards for low-level skill training based on the advantage function of the high-level policy. This auxiliary reward enables efficient, simultaneous learning of the high-level policy and low-level skills without using task-specific knowledge. In addition, we also theoretically prove that optimizing low-level skills with this auxiliary reward will increase the task return for the joint policy. Experimental results show that our algorithm dramatically outperforms other state-of-the-art HRL methods in Mujoco domains[2]. We also find both low-level and high-level policies trained by our algorithm transferable.

## 1  Introduction

Reinforcement Learning (RL) [1] has achieved considerable successes in domains such as games [2, 3] and continuous control for robotics [4, 5]. Learning policies in long-horizon tasks with delayed rewards, such as robot navigation, is one of the major challenges for RL. Hierarchically-structured policies, which allow for control at multiple time scales, have shown their strengths in these challenging tasks [6]. In addition, Hierarchical Reinforcement Learning (HRL) methods also provide a promising way to support low-level skill reuse in transfer learning [7].

The subgoal-based HRL methods have recently been proposed and demonstrated great performance in sparse reward problems, where the high-level policy specifies subgoals for low-level skills to learn [8, 9, 10, 11]. However, the performances of these methods heavily depend on the careful goal space design [12]. Another line of HRL research adopts a pre-training approach that first learns a set of low-level skills with some form of proxy rewards, e.g., by maximizing velocity [13], by designing some simple, atomic tasks [14], or by maximizing entropy-based diversity objective [15]. These methods then proceed to learn a high-level policy to select pre-trained skills in downstream tasks, and each selected skill is executed for a fixed number of steps. However, using fixed pre-trained skills

---

[*]Denotes equal contribution

[2]Videos available at: http://bit.ly/2JxAOeN

without further adaptation is often not sufficient for solving the downstream task, since proxy rewards for pre-training may not be well aligned with the task.

To address these challenges, we develop a novel HRL approach with Advantage-based Auxiliary Rewards (HAAR) to enable concurrent learning of both high-level and low-level policies in continuous control tasks. HAAR specifies auxiliary rewards for low-level skill learning based on the advantage function of the high-level policy, without using domain-specific information. As a result, unlike subgoal-based HRL methods, the low-level skills learned by HAAR are environment-agnostic, enabling their further transfer to similar tasks. In addition, we have also formally shown that the monotonic improvement property of the policy optimization algorithm, such as TRPO [16], is inherited by our method which concurrently optimizes the joint policy of high-level and low-level. To obtain a good estimation of the high-level advantage function at early stages of training, HAAR leverages state-of-the-art skill discovery techniques [13, 14, 15] to provide a useful and diverse initial low-level skill set, with which high-level training is accelerated. Furthermore, to make the best use of low-level skills, HAAR adopts an annealing technique that sets a large execution length at the beginning, and then gradually reduces it for more fine-grained execution.

We compare our method with state-of-the-art HRL algorithms on the benchmarking Mujoco tasks with sparse rewards [17]. Experimental results demonstrate that (1) our method significantly outperforms previous algorithms and our auxiliary rewards help low-level skills adapt to downstream tasks better; (2) annealing skill length accelerates the learning process; and (3) both high-level policy and low-level adapted skills are transferable to new tasks.

## 2    Preliminaries

A reinforcement learning problem can be modeled by a Markov Decision Process (MDP), which consists of a state set $S$, an action set $A$, a transition function $T$, a reward function $R$, and a discount factor $\gamma \in [0, 1]$. A policy $\pi(a|s)$ specifies an action distribution for each state. The state value function $V_\pi(s)$ for policy $\pi$ is the expected return $V_\pi(s) = \mathbb{E}_\pi[\sum_{i=0}^{\infty} \gamma^i r_{t+i} | s_t = s]$. The objective of an RL algorithm in the episodic case is to maximize the $V_\pi(s_0)$, where $s_0 \sim \rho_0(s_0)$.

HRL methods use multiple layers of policies to interact jointly with the environment. We overload all notations in standard RL setting by adding superscripts or subscripts. In a two-layer structure, the joint policy $\pi_{joint}$ is composed of a high-level policy $\pi_h(a^h|s^h)$ and low-level skills $\pi_l(a^l|s^l, a^h)$. Notice that the state spaces of the high level and low level are different. Similar to [13, 18, 19, 20], we factor the state space $S$ into task-independent space $S_{agent}$ and task-related space $S_{rest}$, with which we define $s^l \in S_{agent}$ and $s^h \in S$. The high-level action space in our settings is discrete and $a^h$ is a one-hot vector, which refers to a low-level skill. A low-level skill is a subpolicy conditioned on $a^h$ that alters states in a consistent way [15, 19]. $\gamma_h$ and $\gamma_l$ denote the discount factor for high and low levels, respectively.

## 3    Method

In this section, we present our algorithm, HRL with Advantage function-based Auxiliary Rewards (HAAR). First we describe the framework of our algorithm. Second, we formally define the advantage-based auxiliary reward. Finally, we theoretically prove that the monotonicity of the optimization algorithm used for each level's training is retained for the joint policy.

### 3.1    The HAAR Learning Framework

Figure 1 illustrates the execution cycle of HAAR. At timestep $i$, an agent on state $s_i^h$ takes a high-level action $a_i^h$ encoded by a one-hot vector. $\pi_l$ is a neural network which takes $a_i^h$ and state $s_i^l$ as inputs, and outputs a low-level action $a_i^l$. Different low-level skills are distinguished by different $a_i^h$ injected into this neural network. In this way, a single neural network $\pi_l$ can encode all the low-level skills [13]. The selected low-level skill will be executed for $k$ steps, i.e., $a_j^h = a_i^h (i \le j < i + k)$. After this, the high-level policy outputs a new action. The high-level reward $r_t^h$ is the cumulative $k$-step environment return, i.e., $r_t^h = \sum_{i=0}^{k-1} r_{t+i}$. The low-level reward $r_t^l$ is the auxiliary reward computed from the high-level advantage function, which will be discussed in detail in the next section.

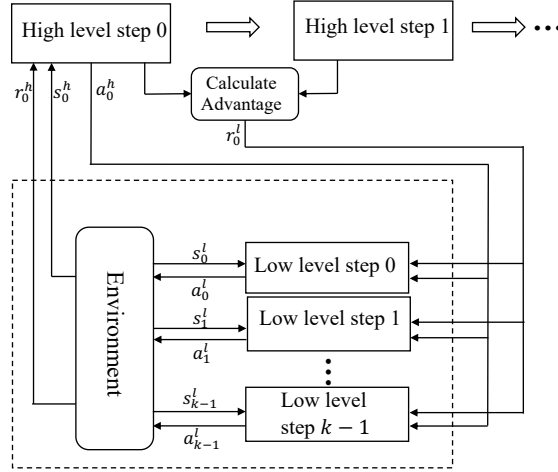

Figure 1: A schematic illustration of HAAR carried out in one high-level step. Within high-level step 0, a total of $k$ low-level steps are taken. Then the process continues to high-level step 1 and everything in the dashed box is repeated.

Algorithm 1 shows the learning procedure of HAAR. In each iteration, we first sample a batch of $T$ low-level time steps by running the joint policy $\pi_{joint}$ in the way shown in Figure 1 (Line 5). Then we calculate the auxiliary reward $r_t^l$ introduced in the next section and replace the environment reward $r_t$ with $r_t^l$ in the low-level experience as $\{s_t^l, a_t^l, r_t^l, s_{t+1}^l\}$ for $0 \leq t < T$ (Line 6). Finally we update $\pi_h$ and $\pi_l$ with Trust Region Policy Optimization (TRPO) [16] using the modified experience of the current iteration (Line 7, 8). In fact, we can use any actor-critic policy gradient algorithm [21] that improves the policy with approximate monotonicity.

In most previous works of skill learning and hierarchical learning [13, 22, 8], the skill length $k$ is fixed. When $k$ is too small, the horizon for the high-level policy will be long and the agent explores slowly. When $k$ is too large, the high-level policy becomes inflexible, hence a non-optimal policy. To balance exploration and optimality, we develop a skill length annealing method (Line 9). The skill length $k_i$ is annealed with the iteration number $i$ as $k_i = k_1 e^{-\tau i}$, where $k_1$ is the initial skill length and $\tau$ is the annealing temperature. We define a shortest length $k_s$ such that when $k_i < k_s$, we set $k_i$ to $k_s$ to prevent the skills from collapsing into a single action.

### 3.2 Advantage Function-Based Auxiliary Reward

The sparse environment rewards alone can hardly provide enough supervision to adapt low-level skills to downstream tasks. Here we utilize the high-level advantage functions to set auxiliary rewards

---

**Algorithm 1** HAAR algorithm

---

1: Pre-train low-level skills $\pi_l$.
2: Initialize high-level policy $\pi_h$ with a random policy.
3: Initialize the initial skill length $k_1$ and the shortest skill length $k_s$.
4: **for** i $\in \{1, ..., N\}$ **do**
5:     Collect experiences following the scheme in Figure 1, under $\pi_h$ and $\pi_l$ for $T$ low-level steps.
6:     Modify low-level experience with auxiliary reward $r_t^l$ defined in Equation (1).
7:     Optimize $\pi_h$ with the high-level experience of the $i$-th iteration.
8:     Optimize $\pi_l$ with the modified low-level experience of the $i$-th iteration.
9:     $k_{i+1} = max(f(k_i), k_s)$.
10: **end for**
11: **return** $\pi_h, \pi_l$.

---

for low-level skills. The advantage function [23] of high-level action $a_t^h$ at state $s_t^h$ is defined as

$$A_h(s^h, a^h) = \mathbb{E}_{s_{t+k}^h \sim (\pi_h, \pi_l)}[r_t^h + \gamma_h V_h(s_{t+k}^h)|a_t^h = a^h, s_t^h = s^h] - V_h(s^h).$$

To encourage the selected low-level skills to reach states with greater values, we set the estimated high-level advantage function as our auxiliary rewards to the low-level skills.

$$R_l^{s_t^h, a_t^h}(s_t^l..s_{t+k-1}^l) = A_h(s_t^h, a_t^h),$$

where $R_l^{s_t^h, a_t^h}(s_t^l..s_{t+k-1}^l)$ denotes the sum of $k$-step auxiliary rewards under the high-level state-action pair $(s_t^h, a_t^h)$. For simplicity, We do a one-step estimation of the advantage function in Equation (1). As the low-level skill is task-agnostic and do not distinguish between high-level states, we split the total auxiliary reward evenly among each low-level step, i.e., we have

$$\begin{aligned} r_i^l &= \frac{1}{k} A_h(s_t^h, a_t^h) \qquad\qquad &(1) \\ {\scriptstyle t \le i < t+k} \\ &= \frac{r_t^h + \gamma_h V_h(s_{t+k}^h) - V_h(s_t^h)}{k}. &(2) \end{aligned}$$

An intuitive interpretation of this auxiliary reward function is that, when the temporally-extended execution of skills quickly backs up the sparse environment rewards to high-level states, we can utilize the high-level value functions to guide the learning of low-level skills.

In order to obtain meaningful high-level advantage functions at early stages of training, we pre-train low-level skills $\pi_l(a_t^l|s_t^l, a_t^h)$ with one of the existing skill discovery algorithms [13, 15] to obtain a diverse initial skill set. This skill set is likely to contain some useful but imperfect skills. With these pre-trained skills, the agent explores more efficiently, which helps the estimate of high-level value functions.

### 3.3   Monotonic Improvement of Joint Policy

In this section we show that HAAR retains the monotonicity of the optimization algorithm used for each level's training, and improves the joint policy monotonically. Notice that in HAAR, low-level skills are optimized w.r.t the objective defined by auxiliary rewards instead of environment rewards. Nevertheless, optimizing this objective will still lead to increase in joint policy objective. This conclusion holds under the condition that (1) the optimization algorithms for both the high and low level guarantee monotonic improvement w.r.t their respective objectives; (2) the algorithm used in the proof is slightly different from Algorithm 1: in one iteration $\pi_l$ is fixed while optimizing $\pi_h$, and vice versa; and (3) discount factors $\gamma_h, \gamma_l$ are close to 1.

We define the expected start state value as our objective function to maximize (for convenience, we use $\pi$ in place of $\pi_{joint}$)

$$\eta(\pi) = \eta(\pi_h, \pi_l) = \mathbb{E}_{(s_t^h, a_t^h) \sim (\pi_h, \pi_l)}\left[ \sum_{t=0,k,2k,...} \gamma_h^{t/k} r_h(s_t^h, a_t^h) \right]. \qquad (3)$$

First, we assert that the optimization of the high level policy $\pi_h$ with fixed low level policy $\pi_l$ leads to improvement in the joint policy. Since the reward for high level policy is also the reward for the joint policy, fixing $\pi_l$ in (3), it essentially becomes the expression for $\eta_h(\pi_h)$. Therefore, $\pi_h$ and $\pi$ share the same objective when we are optimizing the former. Namely, when $\pi_l$ is fixed, maximizing $\eta_h(\pi_h)$ is equivalent to maximizing $\eta(\pi)$.

Now we consider the update for the low-level policy. We can write the objective of the new joint policy $\tilde{\pi}$ in terms of its advantage over $\pi$ as (proved in Lemma 1 in the appendix)

$$\eta(\tilde{\pi}) = \eta(\pi) + \mathbb{E}_{(s_t^h, a_t^h) \sim \tilde{\pi}}\left[ \sum_{t=0,k,2k,...} \gamma_h^{t/k} A_h(s_t^h, a_t^h) \right]. \qquad (4)$$

Since $\eta(\pi)$ is independent of $\eta(\tilde{\pi})$, we can express the optimization of the joint policy as

$$\max_{\tilde{\pi}} \eta(\tilde{\pi}) = \max_{\tilde{\pi}} \mathbb{E}_{(s_t^h, a_t^h) \sim \tilde{\pi}}\left[ \sum_{t=0,k,2k,...} \gamma_h^{t/k} A_h(s_t^h, a_t^h) \right]. \qquad (5)$$

Let $\tilde{\pi}_l$ denote a new low-level policy. In the episodic case, the optimization algorithm for $\tilde{\pi}_l$ tries to maximize

$$\eta_l(\tilde{\pi}_l) = \mathbb{E}_{s_0^l \sim \rho_0^l}[V_l(s_0^l)] = \mathbb{E}_{(s_t^l, a_t^l) \sim (\tilde{\pi}_l, \pi_h)}\left[\sum_{t=0,1,2,\ldots} \gamma_l^t r_l(s_t^l, a_t^l)\right]. \quad (6)$$

Recall our definition of low-level reward in Equation (1) and substitute it into Equation (6), we have (detailed derivation can be found in Lemma 2 in the appendix)

$$\eta_l(\tilde{\pi}_l) = \mathbb{E}_{s_0^l}[V_l(s_0^l)] \approx \frac{1 - \gamma_l^k}{1 - \gamma_l} \mathbb{E}_{(s_t^h, a_t^h) \sim (\tilde{\pi}_l, \pi_h)}\left[\sum_{t=0,k,2k,\ldots} \gamma_h^{t/k} A_h(s_t^h, a_t^h)\right]. \quad (7)$$

Notice how we made an approximation in Equation (7). This approximation is valid when $\gamma_h$ and $\gamma_l$ are close to 1 and $k$ is not exceptionally large (see Lemma 2). These requirements are satisfied in typical scenarios. Now let us compare this objective (7) with the objective in (5). Since $\frac{1 - \gamma_l^k}{1 - \gamma_l}$ is a positive constant, we argue that increasing (7), which is the objective function of the low level policy, will also improve the objective of the joint policy $\pi$ in (5).

In summary, our updating scheme results in monotonic improvement of $\eta(\pi_{joint})$ if we can monotonically improve the high-level policy and the low-level policy with respect to their own objectives. In practice, we use TRPO [16] as our optimization algorithm.

Recall that we make an important assumption at the beginning of the proof. We assume $\pi_h$ is fixed when we optimize $\pi_l$, and $\pi_l$ is fixed when we optimize $\pi_h$. This assumption holds if, with a batch of experience, we optimize either $\pi_h$ or $\pi_l$, but not both. However, this may likely reduce the sample efficiency by half. We find out empirically that optimizing both $\pi_l$ and $\pi_h$ with one batch does not downgrade the performance of our algorithm (see Appendix C.1). In fact, optimizing both policies with one batch is approximately twice as fast as collecting experience and optimizing either policy alternately. Therefore, in the practical HAAR algorithm we optimize both high-level and low-level policies with one batch of experience.

## 4 Related Work

HRL has long been recognized as an effective way to solve long-horizon and sparse reward problems [24, 25, 26, 27]. Recent works have proposed many HRL methods to learn policies for continuous control tasks with sparse rewards [8, 9, 13, 14]. Here we roughly classify these algorithms into two categories. The first category lets a high-level policy *select* a low-level skill to execute [13, 14, 15, 28], which we refer to as the *selector methods*. In the second category, a subgoal is set for the low level by high-level policies [8, 9, 11, 10, 25], which we refer to as *subgoal-based methods*.

Selector methods enable convenient skill transferring and can solve diverse tasks. They often require training of high-level and low-level policies within different environments, where the low-level skills are pre-trained either by proxy reward [13], by maximizing diversity [15], or in designed simple tasks [14]. For hierarchical tasks, low-level skills are frozen and only high-level policies are trained. However, the frozen low-level skills may not be good enough for all future tasks. [22] makes an effort to jointly train high-level and low-level policies, but the high-level training is restricted to a certain number of steps due to approximations made in the optimization algorithm. [13] mentions a potential method to train two levels jointly with a Gumble-Softmax estimator [29]. The Option-Critic algorithm [30] also trains two levels jointly. However, as noted by [11], joint training may lead to loss of semantic meaning of the output of high policies. Therefore, the resulted joint policy in [30] may degenerate into a deterministic policy or a primitive policy (also pointed out in [16]), losing strengths brought by hierarchical structures. To avoid these problems, our algorithm, HAAR, trains both policies concurrently (simultaneously, but in two optimization steps). Furthermore, these joint training algorithms do not work well for tasks with sparse rewards, because training low-level skills requires dense reward signals.

Subgoal-based methods are designed to solve sparse reward problems. A distance measure is required in order for low-level policies to receive internal rewards according to its current state and the subgoal. Many algorithms simply use Euclidean distance [8, 9] or cosine-distance [11] as measurements. However, these distance measure within state space does not necessarily reflect the "true" distance

between two states [31]. Therefore these algorithms are sensitive to state space representation [12]. To resolve this issue, [31] proposes to use actionable state representation, while [32] learns to map the original goal space to a new space with a neural network. Our algorithm HAAR, on the other hand, manages to avoid the thorny problem of finding a proper state representation. By using only the advantage function for reward calculation, HAAR remains domain-independent and works under *any* state representation.

The way we set auxiliary rewards share some similarities with the potential-based reward shaping methods [33], which relies on heuristic knowledge to design a potential reward function to facilitate the learning. In contrast, our method requires no prior knowledge and is able to take advantage of the hierarchical structure.

# 5 Experiments

## 5.1 Environment Setup

We adapt the benchmarking hierarchical tasks introduced in [17] to test HAAR. We design the observation space such that the low-level skills are task-agnostic, while the high-level policy is as general as possible. The low level only has access to the agent's joint angles, stored in $s^l$. This choice of low-level observation necessitates minimal domain knowledge in the pre-training phase, such that the skill can be transferred to a diverse set of domains. This is also discussed in [13]. The high level can perceive the walls/goals/other objects by means of seeing through a range sensor - 20 "rays" originating from the agent, apart from knowing its own joint angles, all this information being concatenated into $s^h$. To distinguish between states, the goal can always be seen regardless of walls.

Note that unlike most other experiments, the agent does not have access to any information that directly reveals its absolute coordinates ($x, y$ coordinates or top-down view, as commonly used in HRL research experiments). This makes our tasks more challenging, but alleviates over-fitting of the environment and introduces potential transferability to both $\pi_h$ and $\pi_l$, which we will detail on later. We compare our algorithm to prior methods in the following tasks:

- Ant Maze: The ant is rewarded for reaching the specified position in a maze shown in Figure 2(a). We randomize the start position of the ant to acquire even sampling of states.
- Swimmer Maze: The swimmer is rewarded for reaching the goal position in a maze shown in Figure 2(b).
- Ant Gather: The ant is rewarded for collecting the food distributed in a finite area while punished for touching the bombs, as shown in Figure 2(c).

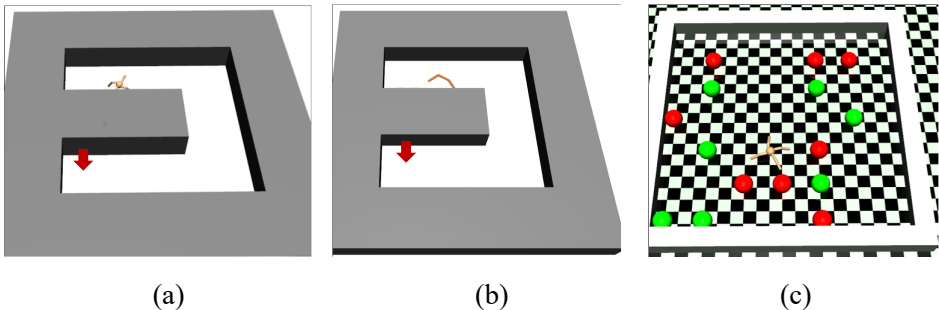

(a)          (b)          (c)

Figure 2: A collection of environments that we use. (a) Ant in maze (b) Swimmer in maze (c) Ant in gathering task.

## 5.2 Results and Comparisons

We compare our algorithm with the state-of-the-art HRL method SNN4HRL [13], subgoal-based methods HAC [8] and HIRO [9] and non-hierarchical method TRPO [16] in the tasks above. HAAR and SNN4HRL share the same set of pre-trained low-level skills with stochastic neural network.

HAAR significantly outperforms baseline methods. Some of the results are shown in Figure 3. All curves are averaged over 5 runs and the shaded error bars represent a confidence interval of 95%. In all the experiments, we include a separate learning curve of HAAR without annealing the low-level skill length, to study the effect of annealing on training. Our full implementation details are available in Appendix B.

**Comparison with SNN4HRL and Non-Hierarchical Algorithm**

Compared with SNN4HRL, HAAR is able to learn faster and achieve higher convergence values in all the tasks[3]. This verifies that mere pre-trained low-level skills are not sufficient for the hierarchical tasks.

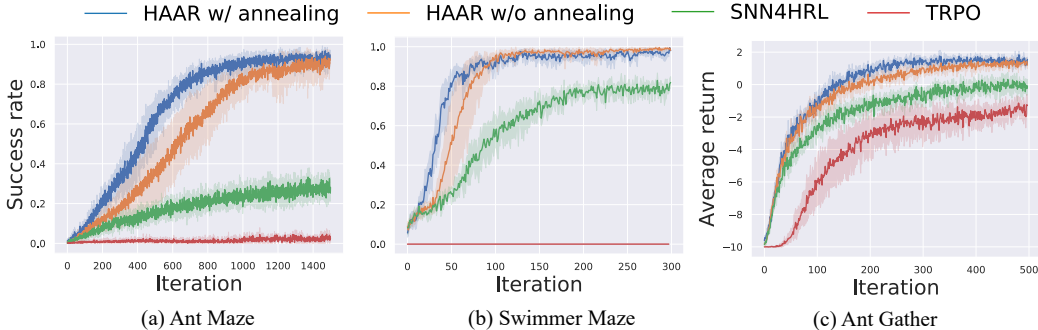

(a) Ant Maze     (b) Swimmer Maze     (c) Ant Gather

Figure 3: Learning curves of success rate or average return in Ant Maze, Swimmer Maze and Ant Gather tasks. The curves are HAAR with skill annealing, HAAR without skill length annealing, SNN4HRL and TRPO, respectively.

The success rate of SNN4HRL in the Swimmer Maze task is higher than that of the Ant Maze task because the swimmer will not trip over even if low-level skills are not fine-tuned. Nevertheless, in Swimmer Maze, our method HAAR still outperforms SNN4HRL. HAAR reaches a success rate of almost 100% after fewer than 200 iterations.

The main challenges of Ant Gather task is not sparse rewards, but rather the complexity of the problem, as rewards in the Ant Gather task is much denser compared to the Maze environment. Nevertheless, HAAR still achieves better results than benchmark algorithms. This indicates that HAAR, though originally designed for sparse reward tasks, can also be applied in other scenarios.

TRPO is non-hierarchical and not aimed for long-horizon sparse reward problems. The success rates of TRPO in all maze tasks are almost zero. In Ant Gather task, the average return for TRPO has a rise because the ant robot learns to stay static and not fall over due to the death reward $-10$.

**Comparison with Subgoal-Based Methods**

We also compare our method HAAR with the state-of-the-art subgoal-based HRL methods, HAC and HIRO in the Ant Maze environment. Because we use a realistic range sensor-based observation and exclude the $x, y$ coordinates from the robot observation, subgoal-based algorithms cannot properly calculate the distances between states, and perform just like the non-hierarchical method TRPO. We even simplify the Maze task by placing the goal directly in front of the ant, but it is still hard for those subgoal-based methods to learn the low-level gait skills. Therefore, we omit them from the results.

This result accords with our previous analysis of subgoal-based algorithms and is also validated by detailed studies in [12], which mutates state space representation less than we do, and still achieves poor performance with those algorithms.

**The Effect of Skill Length Annealing**

HAAR without annealing adopts the same skill length as HAAR with annealing at the end of training, so that the final joint policies of two training schemes are the same in structure. The learning curves are presented in Figure 3. In general, training with skill length annealing helps the agent learn faster. Also, annealing has no notable effect on the final outcome of training, because the final policies, with or without annealing, share the same skill length $k$ eventually. We offered an explanation for this effect at the end of the Section 3.1.

## 5.3 Visualization of Skills and Trajectories

To demonstrate how HAAR is able to achieve such an outperformance compared to other state-of-the-art HRL methods, we provide a deeper look into the experimental results above. In Figure 4, we compare the low-level skills before and after training in the Ant Maze task.

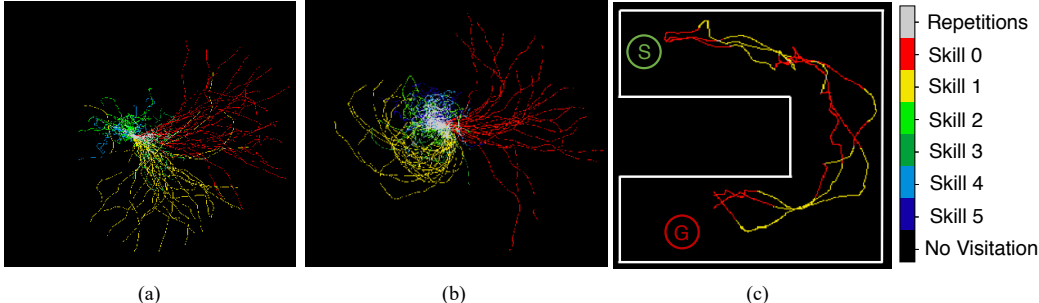

| | |
|---|---|
| | Repetitions |
| | Skill 0 |
| | Skill 1 |
| | Skill 2 |
| | Skill 3 |
| | Skill 4 |
| | Skill 5 |
| | No Visitation |

(a)　　　　　　　　　(b)　　　　　　　　　(c)

Figure 4: (a) Visitation plot of initial low-level skills of the ant. (b) Low-level skills after training with auxiliary rewards in Ant Maze. (c) Sample trajectories of the ant after training with HAAR in Ant Maze.

In Figure 4, (a) and (b) demonstrate a batch of experiences collected with low-level skills before and after training, respectively. The ant is always initialized at the center and uses a single skill to walk for an arbitrary length of time. Comparing (b) with (a), we note that the ant learns to turn right (Skill 1 in yellow) and go forward (Skill 0 in red) and well utilizes these two skills in the Maze task in (c), where it tries to go to (G) from (S). We offer analysis for other experiments in Appendix C.

In our framework, we make no assumption on how those initial skills are trained, and our main contribution lies in the design of auxiliary rewards to facilitate low-level control training.

## 5.4 Transfer of Policies

Interestingly, even though HAAR is not originally designed for transfer learning, we find out in experiments that both $\pi_h$ and $\pi_l$ could be transferred to similar new tasks. In this section we analyze the underlying reasons of our method's transferability.

We use the Ant Maze task shown in Figure 2(a) as the source task, and design two target tasks in Figure 5 that are similar to the source task. Target task (a) uses a mirrored maze (as opposed to the original maze in Figure 2(a)) and target task (b) is a spiral maze. Now we test the transferability of HAAR by comparing the learning curves of (1) transferring both high-level and low-level policies, (2) transferring the low-level alone, and (3) not transferring any policy. We randomly pick a trained $\pi_l$ and its corresponding $\pi_h$ from the learned policies in the experiment shown in Figure 2(a), and apply them directly on tasks in Figure 5.

HAAR makes no assumption on state space representation. Therefore, in experiments we only allow agents access to information that is universal across similar tasks. First, as defined in Section 2, $s^l$ is the ego-observation of the agent's joint angles. This ensures the low-level skills are unaware of its surroundings, hence limited to atomic actions and avoids the problem of the joint policy being degenerated to always using the same low-level policy [11, 30].

Apart from information in $s^l$, the high level can also perceive surrounding objects through a range sensor. We block its access to absolute coordinates so that the agent does not simply remember the environment, but learns to generalize from observation. Our experimental results in Figure 5 (c)(d)

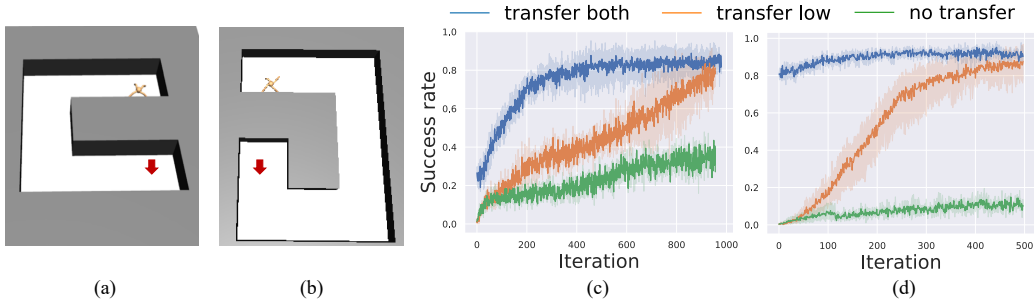

Figure 5: (a) and (b) are tasks to test the transferability of learned policies. (c) and (d) are the corresponding learning curves of transferring both high and low-level policies, transferring only low-level policy, and not transferring (the raw form of HAAR).

verifies that both low-level and high-level policies are indeed transferable and can facilitate learning in a similar new task.

For both target tasks, there is a jump start of success rate by transferring both $\pi_h$ and $\pi_l$. The learning curve of target task (b) enjoys a very high jump start due to its trajectorical similarity to the source task. Transferring both $\pi_h$ and $\pi_l$ results in very fast convergence to optimum. Transferring only $\pi_l$ also results in significantly faster training compared to non-transfer learning. This indicates that with HAAR, the agent learns meaningful skills in the source task (which is also analyzed by Figure 4). By contrast, it is unlikely for works that rely on coordinates as part of the observation, such as [9], to transfer their policies.

We want to point out that as the maze we use in this experiment is simple, the agent could possibly derive its location according to its observation, therefore still over-fitting the environment to some extent. However, using more complex mazes as source tasks may resolve this problem.

## 5.5 Discussion of State Observability

In our experiments, the decision process on the high level is clearly an MDP since states are definitive from the observation. We notice that the low level states, however, are not fully observable. Direct information about the maze(walls and the goal) is excluded from the low level. Nevertheless, indirect information about the maze is expressed through $a_h$, which is a function of wall and goal observation. Strictly speaking, the low-level dynamics is a partially observable Markov decision process. But owing to the indirect information carried in $a_h$, we use $s_l$ to approximate the complete state and still apply TRPO on it. Experimental results verify the validity of such approximation. This approximation could be avoided by taking partial observability into consideration. For example, the GTRPO algorithm [34] can be utilized to optimize the low-level policy.

## 6 Conclusion

In this work, we propose a novel hierarchical reinforcement learning framework, HAAR. We design a concurrent training approach for high-level and low-level policies, where both levels utilize the same batch of experience to optimize different objective functions, forming an improving joint policy. To facilitate low-level training, we design a general auxiliary reward that is dependent only on the high-level advantage function. We also discuss the transferability of trained policies under our framework, and to combine this method with transfer learning might be an interesting topic for future research. Finally, as we use TRPO for on-policy training, sample efficiency is not very high and computing power becomes a major bottleneck for our algorithm on very complex environments. To combine off-policy training with our hierarchical structure may have the potential to boost sample efficiency. As the low-level skill initialization scheme has a dramatic influence on performance, an exploration of which low-level skill initialization scheme works best is a future direction as well.

**Acknowledgments**

The authors would like to thank the anonymous reviewers for their valuable comments and helpful suggestions. The work is supported by Huawei Noah's Ark Lab under Grant No. YBN2018055043.

## Footnotes

[3]In our comparative experiments, the numbers of timesteps per iteration when training with SNN4HRL is different from that in the original paper [13]. SNN4HRL's performance, in terms of timesteps, is consistent with the original paper.

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
