[Supplementary Material · Appendix_HAAR.pdf]

# A   Additional Proof

**Lemma 1** *The objective of a new joint policy $\tilde{\pi}$ can be written in terms of the advantage over $\pi$ as*

$$\eta(\tilde{\pi}) = \eta(\pi) + \mathbb{E}_{(s_t^h, a_t^h) \sim \tilde{\pi}}\left[\sum_{t=0,k,2k,\dots} \gamma_h^{t/k} A_h(s_t^h, a_t^h)\right] \tag{8}$$

**Proof A.1** *For an objective function defined as*

$$\begin{aligned}
\eta(\pi) &= \mathbb{E}_{s_0^h}[V_h(s_0^h)] \\
&= \mathbb{E}_{s_0^h, a_0^h, \dots \sim \pi}\left[\sum_{t=0,k,2k,\dots} \gamma_h^{t/k} r_h(s_t^h)\right]
\end{aligned} \tag{9}$$

*Changing $\pi$ to $\tilde{\pi}$, we have*

$$\begin{aligned}
&\mathbb{E}_{s_0^h, a_0^h, \dots \sim \tilde{\pi}}\left[\sum_{t=0,k,2k,\dots} \gamma_h^{t/k} A_h(s_t^h, a_t^h)\right] \\
&= \mathbb{E}_{s_0^h, a_0^h, \dots \sim \tilde{\pi}}\left[\sum_{t=0,k,2k,\dots} \gamma_h^{t/k} (r_h(s_t^h) + \gamma_h V_h(s_{t+k}^h) - V_h(s_t^h))\right] \\
&= \mathbb{E}_{s_0^h, a_0^h, \dots \sim \tilde{\pi}}\left[-V_h(s_0^h) + \sum_{t=0,k,2k,\dots} \gamma_h^{t/k} r_h(s_t^h)\right] \\
&= -\eta(\pi) + \eta(\tilde{\pi})
\end{aligned} \tag{10}$$

*Rearranging two sides of the equation, we obtain*

$$\eta(\tilde{\pi}) = \eta(\pi) + \mathbb{E}_{(s_t^h, a_t^h) \sim \tilde{\pi}}\left[\sum_{t=0,k,2k,\dots} \gamma_h^{t/k} A_h(s_t^h, a_t^h)\right] \tag{11}$$

**Lemma 2** *With the auxiliary reward of $\underset{t \le i < t+k}{r_i^l} = \frac{1}{k} A_h(s_t^h, a_t^h)$, the low level policy $\tilde{\pi}_l$'s expected start value can be written as*

$$\mathbb{E}_{s_0^l}[V_l(s_0^l)] \approx \frac{1 - \gamma_l^k}{1 - \gamma_l} \mathbb{E}_{\tau_h \sim (\tilde{\pi}_l, \pi_h)}\left[\sum_{t=0,k,2k,\dots} \gamma_h^{t/k} A_h(s_t^h, a_t^h)\right] \tag{12}$$

*Where $\tau_h$ is the trajectory of high-level steps $= s_0^h, a_0^h, s_k^h, a_k^h, \dots$. The approximation is correct under the condition that the fixed high level policy $\pi_h$, the low level discount factor $\gamma_l$ as well as the high level discount factor $\gamma_h$ are both close to 1, and the skill length $k$ is not extremely large.*

**Proof A.2** *We define $\tau$ to be the trajectory sampled with $(\tilde{\pi}_l, \pi_h)$.*

$$\tau = s_0^h, a_0^h, s_1^l, a_1^l, \dots, s_{nk}^h, a_{nk}^h, s_{nk+1}^l, a_{nk+1}^l \dots$$

*The high-level trajectory is defined as*

$$\tau_h = s_0^h, a_0^h, s_k^h, a_k^h, \dots, s_{nk}^h, a_{nk}^h, \dots$$

*The k-step low level trajectory within a single step of $(s_t^h, a_t^h)$ is defined as*

$$\tau_l(t) = s_t^l, a_t^l, \dots s_{t+k-1}^l, a_{t+k-1}^l$$

*Now we can write our low level policy's expected start value as*

$$\mathbb{E}_{s_0^l}[V_l(s_0^l)] = \mathbb{E}_{\tau \sim (\tilde{\pi}_l, \pi_h)}\left[\sum_{t=0,1,2,\dots} \gamma_l^t r_l(s_t^l, a_t^l)\right] \tag{13}$$

$$= \mathbb{E}_{\tau_h \sim (\tilde{\pi}_l, \pi_h)}\left[\sum_{t=0,k,2k,\dots} \mathbb{E}_{\tau_l(t) \sim (\tilde{\pi}_l, \pi_h)}\left[\sum_{i=0}^{k-1} \gamma_l^{t+i} A_h(s_t^h, a_t^h)\right]\right] \tag{14}$$

$$= \mathbb{E}_{\tau_h \sim (\tilde{\pi}_l, \pi_h)}\left[\sum_{t=0,k,2k,\dots} \sum_{i=0}^{k-1} \gamma_l^{t+i} A_h(s_t^h, a_t^h)\right] \tag{15}$$

$$= \mathbb{E}_{\tau_h \sim (\tilde{\pi}_l, \pi_h)}\left[\sum_{t=0,k,2k,\dots} \gamma_l^t \frac{1-\gamma_l^k}{1-\gamma_l} A_h(s_t^h, a_t^h)\right] \tag{16}$$

$$= \frac{1-\gamma_l^k}{1-\gamma_l} \mathbb{E}_{\tau_h \sim (\tilde{\pi}_l, \pi_h)}\left[\sum_{t=0,k,2k,\dots} \gamma_l^t A_h(s_t^h, a_t^h)\right] \tag{17}$$

*When $\gamma_l$ and $\gamma_h$ are both close to 1 and $k$ is not exceedingly large, we can approximate $\gamma_l$ with $\gamma_h^{1/k}$ like below*

$$\frac{1-\gamma_l^k}{1-\gamma_l} \mathbb{E}_{\tau_h \sim (\tilde{\pi}_l, \pi_h)}\left[\sum_{t=0,k,2k,\dots} \gamma_l^t A_h(s_t^h, a_t^h)\right]$$

$$\approx \frac{1-\gamma_l^k}{1-\gamma_l} \mathbb{E}_{\tau_h \sim (\tilde{\pi}_l, \pi_h)}\left[\sum_{t=0,k,2k,\dots} \gamma_h^{t/k} A_h(s_t^h, a_t^h)\right] \tag{18}$$

## B  Implementation Details

### B.1  Experiments Details

Parameters that need to be set for these experiments and that would potentially influence the results of our training include number of iterations $N$ used for the pre-training of low-level skills in with SNN4HRL[13], total low-level step number for experience collection $B$, discount factor for high level $\gamma_h$, discount factor for low level $\gamma_l$, maximal time steps within an episode $T$, number of low steps within a high step $k_s$ (in the none-annealing case) and initial number of low steps within a high step $k_0$ (in the annealing case). Note that we always define the annealing temperature such that the skill length is fully annealed and no longer changes half way through training (around $\frac{B}{2}$ low steps). We did not perform a grid search on hyperparameters, therefore better performances might be possible for these experiments.

We use the ant agent pre-defined in rllab. The observation of high-level $s^h$ include its ego-observation (joint angles and speed), as well as the perception of walls and goals through a 20-ray range sensor. The observation of low-level $s^l$ consists only of the agent's ego observations.

| Hyperparameters for experiments | | | |
|---|---|---|---|
| Hyperparameter | Ant Maze | Swimmer Maze | Ant Gather |
| N | 1000 | 500 | 3000 |
| B | $5 \times 10^4$ | $5 \times 10^5$ | $5 \times 10^4$ |
| $\gamma_l$ | 0.99 | 0.99 | 0.99 |
| $\gamma_h$ | 0.99 | 0.99 | 0.99 |
| $k_0$ | 100 | 1000 | 100 |
| $k_s$ | 10 | 500 | 10 |
| $T$ | 1000 | 5000 | 1000 |

**Ant Maze** A "C"-shaped maze is constructed with multiple $4 \times 4$ blocks or empty space. The episode terminates when the ant reaches the goal (with a positive reward 1000), runs out of the maximal number of steps, or trips over (with a negative reward $-10$). The goal is placed with in a $4 \times 4$ box and we determine the agent has reached the goal once its center of mass is within this box.

**Swimmer Maze** Swimmer Maze uses the same maze as the one defined in Ant Maze. The episode terminates when the swimmer reaches the goal (with a positive reward 1000) or runs out of the maximal number of steps. The swimmer does not trip over.

**Ant Gather** Ant Gather uses a $6 \times 6$ maze with 8 randomly generated food items and 8 randomly generated bombs. If the ant gets a food item it will receive a reward of 1. If it reaches a bomb it will receive a reward of $-1$. The tripping penalty is $-10$.

### B.2 Training Details

We use perceptron networks with 2 layers of 32 hidden units for $\pi_h$ and $\pi_l$. For the value functions $V(s)$, we use polynomial estimators $V(s) = W_3 s^3 + W_2 s^2 + W_1 s + W_0$. The step size for TRPO is 0.01.

### B.3 Pre-training

The number of skills pre-trained is 6. Maximum path length for ant is $50,000$ and for swimmer is $500$. The hyperparameter $\alpha_H$ defined in SNN4HRL, which is a coefficient used for the MI bonus, is set to 1.

### B.4 Transfer Learning

There is no annealing in both transfer experiments, and the skill length is always 10. For transfer task (a) (mirrored maze), hyperparameters are the same as those used in Ant Maze. For transfer task (b) (spiral maze), $B = 5 \times 10^5$ and $T = 1300$. Hyperparameters not mentioned here are defaulted to the same hyperparmeters used in Ant Maze.

## C   Additional Experiment Results

### C.1   Concurrent Optimization versus Alternate Optimization

Figure 6: Comparison between the concurrent optimization and alternate optimization scheme in Ant Maze task.

Recall that we make an important assumption at the beginning of the proof. We assume $\pi_h$ is fixed when we optimize $\pi_l$, and $\pi_l$ is fixed when we optimize $\pi_h$. This assumption holds if, with a batch of experience, we optimize either $\pi_h$ or $\pi_l$, but not both. One example of this training scheme is to collect one batch of experience and optimize $\pi_h$, then collect another batch and optimize $\pi_l$, and repeatedly carry out this procedure.

However, in practice, we optimize both $\pi_h$ and $\pi_l$ *concurrently* with a single batch of collected experience. To study the effect of this approximation on our training performance, we compare the training curves of our method with the rigorous method which trains two policies *alternately*.

As is shown in Figure 6, optimizing both policies concurrently is approximately twice as fast as optimizing either policy alternately. The choice of optimization scheme does not affect the convergence value of final success rate. Therefore, concurrent training does not downgrade the performance, and it is safe to use it for higher sample efficiency.

## C.2 Analysis of Swimmer Maze

Similar to the Ant Maze task, the agent in Swimmer Maze task also demonstrates meaningful low-level skills after training. In Figure 7(a)(b), the swimmer is initialized at the center of the area and placed horizontally. We plot a set of trajectories of the swimmer resulted by carrying out a single skill for certain amount of steps. (c) demonstrates several trajectories of the swimmer in the task, going from start (S) to goal (G). We can see that the swimmer learns to modify its skills for more effective movement.

Figure 7: Illustration of the swimmer's skills. (a) visitation plot of the swimmer's skills before training (b) visitation plot of swimmer's skills after training (c) several trajectories of the swimmer in the maze.

## C.3 Starting with Random Low-Level Skills

We initialize the low-level skills with random policies and compare with SNN4HRL in the Ant Maze task. Even with random low-level skill initialization, HAAR outperforms SNN4HRL, shown in Figure 8 below. More reasonable initialization results in better performance. We will explore the effects of other initialization schemes in future works.

Figure 8: Comparison with SNN4HRL when starting with random low-level skills in the Ant Maze task.