[Reviews · NeurIPS 2019]

Reviewer 1



The authors present a new hierarchical reinforcement learning (HRL) algorithm that outperforms other state-of-the-art HRL algorithms. The algorithm starts with pre-trained low-level skills provided by existing methods from other hierarchical approaches. The HAAR approach presented in this paper then jointly optimizes a high-level selector policy and adjusts the pre-trained low-level skill policies. The key being that the low-level skill updates are performed using a proxy reward based on the advantage achieved in the high level policy from executing this skill. The authors give a theoretical analysis that such an approach leads to a converging policy when learned with TRPO. Empirically the authors compare to a number of existing HRL baselines, and non-hierarchical TRPO on multiple continuous control domains. I think this is a good approach with good theoretical justification. Experiments support the method, showing it is faster. The empirical analysis is also well done. Minor typo in algorithm 1 line 3 "shorest" -> shortest I've amended my score after the author rebuttals and discussion with the other reviewers. The major concern being about the markovian assumptions with the experiments/representation.

Reviewer 2



## Approach and methodology The main contribution is based around the use of the advantage signal to train the lower level controller. This is an interesting approach and seems novel in the context of options, although it looks to have some similarities to potential based reward shaping, e.g. (Devlin and Kudenko, 2012). The main advantages claimed for HAAR are (loosely) those of improved performance under sparse rewards and the learning of skills appropriate for transfer. These claims could be made more explicit, and that might help to justify the experimental section. The authors define advantage as: $$A_h(s_t^h,a_t^h) = E[r_t^h + \gamma_h V_h(s_{t+k}^h) - V_h(s_{t}^h)]$$ The meaning of this is a little ambiguous and I would prefer this to be clarified. Conventionally, it would be that the advantage of some action a in state s was $$A(s,a) = Q(s,a) - V(s)$$ The Q and V functions are expectations with different conditions on them, so in the context of HAAR, this would amount so something like: $$A_h(s,a) = E[r_t^h + \gamma_h V_h(s_{t+k}^h)| a_{t}^h=a ,s_{t}^h=s] - V_h(s)$$ The authors refer to skills but without a working definition of what this means. I found this (Eysenbach et al., 2018) > A skill is a latent-conditioned policy that alters that state of the environment in a consistent way. I can trace the use of the term in this context back to (Konidaris and Barto, 2007). The authors divide HRL into two general approaches: > The first category lets a high-level policy select a low-level skill to execute [15, 13, 14, 27], which we refer to as the selector methods. In the second category, a subgoal is set for the low level by high-level policies [23, 10, 9, 8, 11], which we refer to as subgoal-based methods. Is this their own categorisation or does it derive from earlier work. If it is their own work, it is a meaningful theoretical contribution and more could be made of it. Later they say that: > Subgoal-based methods are designed to solve sparse reward problems. But they don't really say what fundamental problem selector based methods are there to solve. Is it for transfer or for sparse rewards or something else? When they say rather disengenuous to say: > unlike subgoal-based HRL methods, the low-level skills learned by HAAR are environment-agnostic... This is really true of selector methods that learn low level skills jointly with high level control, isn't it? E.g. SNN4HRL, Option-Critic? ## Monotonic improvement The monotonic improvement result is not laid out as clearly as it could be. I would like to know all the assumptions that are being made (e.g. does it rely on the Markov-ness of the states, both high and low level) [lines 146-7] Contain a claim that is not appropriately justified. It is based on limited experimentation rather than formally defined. > Therefore, we argue that our algorithm retains the property of (approximately) monotonic improvement. This is not demonstrated theoretically, but based on a few empirical evaluations. Arguably, to show this formally you may have to consider learning at two time-scales. ## Related work They authors state [lines 149-50] > HRL has long been recognized as an effective way to solve long-horizon and sparse reward problems [citing work from 1990s and early 2000s] That was the claim but (arguably) it wasn't really demonstrated effectively then, rather it was posited. More recent work has begun to demonstrate this more convincingly. ## Experiments The experiments appear to be challenging and diverse, but the authors could do more to motivate why they are evaluating in each case. If it is a measure of sparseness of reward then perhaps different degrees of sparsity could be explored. ### Annealing They say [line 48]: > annealing skill length greatly accelerates the learning process But the experimental results in Figure 3, show that the difference between annealing and not is variable, and for some tasks (ant gather) of marginal benefit. Perhaps the claim should be more appropriately pitched. ### State representations The identification of transferable representations of states is not a new one. It can be seen as early as (Konidaris and Barto 2007) and an appropriate source should be cited for these ideas. [lines 184-5] contain the lines: > The high level can perceive the walls/goals/other objects by means of seeing through a range sensor - 20 “rays” originating from the agent, apart from knowing its own joint angles, all this information being concatenated into [the high level state observation] This is also a transferable representation of state, but has been excluded from the low level controller. What are the criteria on which these decisions were made? On [line 224-5] they say: > Because we use a realistic range sensor-based observation of states and exclude the x, y coordinates from the robot observation, subgoal-based algorithms cannot properly calculate the distances between states It should be noted that this effectively destroys the Markov nature of the state, and hence the distance between states measure can incorrectly assume as state is very similar (or the same) when it is arbitrarily different. It is still unclear whether the authors' theoretical results rely on states being markov, if not it may be that a different non-markov representation could be manufactured that caused HAAR to underperform significantly. ## Transfer learning They say: > Transferring only $\pi_l$ also results in much faster training It would be good to have a measure of transfer here, e.g. area under the learning curve (see (Konidaris, 2007) or similar). But by any such measure, I don't think just transfering the low level policy is training *much faster*, maybe a little. The major improvement comes from transfering both low and high level, which suggests that the tasks are significantly similar (if there is a good deal of information in the high level policy). Moreover, it isn't clear whether this improvement comes by virtue of the transferable state representation (something identified as early as 2007 in (Konidaris, 2007)) or whether this is a virtue of the newly proposed HAAR algorithm. The SNN4HRL algorithm and paper used in the experiments shows roughly 600 iterations before attaining a 0.2 success rate on average. However, the original paper appears to use the same (or a very similar) experimental set up but achieves a 0.2 mean success rate after approximately 10 iterations and 0.8 after 25 or so. The authors should make it clear what the differences are (if any) that explain such a different (a factor of 60). Does this also relate to the state representation? If so, this should be stated. ### After reading author rebuttal and discussions with other reviewers I agree with the other reviewers that this is a really interesting idea and were the issues I raise dealt with in a future version of the paper, I would recommend accept for that paper. However, as things stand, I do not feel that the authors have rebutted the concerns about the current version of the paper with respect to the theoretical result (namely the claim of approximate monotonic improvement) and the experimental shortcomings (fair comparison which challenges a broad set of non-Markovian properties and improved transfer experiments). I feel that my other concerns were appropriately addressed in the rebuttal (but I would suggest revising the advantage function equation in line with what the authors indicate in the rebuttal). However, my assessment is made on the state of the paper as it was originally submitted, and, for the reasons given above, I am still recommending reject. If the paper is not accepted, I recommend the authors revise and submit again to this or a similar conference as the work is of interest to me and others in the community.

Reviewer 3



The paper is well written and organized. The contribution, a scheme for hierarchical RL (HRL) in the online, on-policy setting, where the low-level policies are adapted, using the advantage function of the high level policy, seams to be novel, is elegant and might be a real improvement for HRL. The idea to improve the method by starting with large time steps and decrease them in an annealing like fashion, is probably a good idea in this kind of algorithms. AFTER FEEDBACK The authors answer to my first question "How will the algorithm perform when starting with random low-level policies?" is very convincing. I found the answer to my second question on the Markov property not very informative, and the authors did not explain, what is meant by the "low-level skills" are "environment-agnostic" and "task-agnostic". I did not change the "overall score".

[Author Response · NeurIPS 2019]

**Response to Reviewer 1:** Thank you for the thoughtful and inspiring comments.

**Q1.** How much of an effect does low-level skill initialization scheme have on performance?

We test random skill initialization in Ant Maze task. Even random initialization is better than the non-hierarchical
method TRPO, shown in Fig.5 below. And more reasonable initialization results in better performance. We will explore
the effects of other initialization schemes in future works.

**Response to Reviewer 2:** Thank you for the detailed comments.

**Q1.** Consider potential based reward shaping in main evaluation.

We design a heuristic potential as the negative L2 distance between agent
and goal. The curve of potential reward is higher than TRPO, but significantly
worse than HAAR, shown in Fig.5. We note that the potential reward shaping
method could not take advantage of hierarchical structure; it also heavily depends on the potential function design.

Figure 1: Task 1     Figure 2: Task 2

**Q2.** More robust and systematic transfer experiments.

We design more new tasks in Fig.1(bigger maze), 2(sprial maze) to explore the effectiveness of low-level skill transfer.
In the new tasks, the skill of turning right learned in Ant Maze (a) can be very useful, so low-level policy transfer shows
much efficiency (shown in Fig.3, 4). New tasks and the old task share much information on the high level, so "transfer
both" performs well, which partly comes by virtue of the transferable state representation (Konidaris, 2007).

Figure 3: Transfer results for task 1.     Figure 4: Transfer results for task 2.     Figure 5: Potential based reward shaping and random low-level skill initialization in Ant Maze.

**Q3.** All assumptions for the proof of monotonic improvement.

(1) All assumptions for TRPO, including that the high-level and low-level states are Markovian; (2)The high level
policy is fixed while optimizing the low level policy and vice versa [lines 116-7]; (3) Discount factors $\gamma_h \to 1, \gamma_l \to 1$
[lines 132-3].

**Q4.** Experiment setting seems to be non-Markovian [line 224-5], different states may have very similar representation.

In experiment, the agent uses a total of 20 rays to "see" the surroundings, and the goal can always be seen regardless of
walls. We believe this is sufficient to distinguish between states, so it is approximately Markovian.

**Q5.** The benchmark, SNN4HRL, seems to run much faster in its original paper compared to in this paper.

The reviewer may have misread the experiment settings. Our result is actually consistent with the original SNN4HRL
paper. In Swimmer Maze, the numbers of samples per iteration are different in SNN4HRL paper and our paper [line
405]. The performance in terms of samples is consistent. For Ant Maze, SNN4HRL paper does not provide results.

**Q6.** A precise description of the advantage function.

Our definition of advantage is consistent with the conventional definition, $Q(s,a) - V(s)$. Using a one-step expansion
of $Q$, we can write it as $A_h(s^h) = E_{s^h_{t+k} \sim (\pi_h, \pi_l)}[r^h_t + \gamma_h V_h(s^h_{t+k})|a^h_t = a^h, s^h_t = s^h] - V_h(s^h)$.

**Q7.** How to determine what to include in low-level state in experiments?

Our decision of what is included in low-level state is the same as SNN4HRL paper (such that the representation requires
minimal domain knowledge in the pre-training phase), described in "Problem Statement" of their paper.

**Response to Reviewer 3:** Thank you for the thoughtful comments.

**Q1.** How will the algorithm perform when starting with random low-level policies?

We run experiments with random initial low-level policies in Ant Maze. Results in Fig.5 show that it performs better
than TRPO. As expected, more meaningful low-level policies result in better performance.

**Q2.** Low level Markovness is not clear. Discuss the Markovness of states.

States for both high level and low level are Markovian. We concatenate the agent state and the high-level action $a^h$ as
the low-level state, so low-level policy is still running on an MDP.

[Meta-Review · NeurIPS 2019]

The paper presents HAAR - a hierarchical reinforcement learning approach that is based on the idea of using the advantage / temporal difference error of the high-level controler provide the reward signal for the lower layer. The reviewers judged this approach to be novel, and empirical results are promising. Analytical results provide improvement guarantees similar to a base algorithm like TRPO. Several areas for improvement were mentioned, and many of these were addressed in the rebuttal. For example, the reviewers were pleased to see the additional experiment showing performance from random skill initialization. Remaining questions after the rebuttal were as follows. First, it was not clear to what the approach may require full observability, and whether the present experiments were specifically constructed with this in mind. A clear specification of each observation space should be provided in the camera ready version, and limitations of the approach (e.g., in terms of partial observability) should be discussed. In addition, there are remaining questions about the precise assumptions underlying the presented analysis. The current claim is too broad, as it is not qualified by the specific assumptions made. Overall, the paper is judged to make a valuable contribution. I urge the authors to carefully consider all reviewer suggestions to improve the camera ready version of the paper.